# REPRESENTATION MATCHING INFORMATION BOTTLE-NECK FOR TEXT MATCHING IN ASYMMETRICAL DO-MAINS

## ABSTRACT

Recent studies have shown that the domain matching of text representations will help improve the generalization ability of asymmetrical domains text matching tasks. This requires that the distribution of text representations should be as similar as possible, similar to matching with heterogeneous data domains, in order to make the data after feature extraction indistinguishable. However, how to align the distribution of text representations remains an open question, and the role of text representations distribution alignment is still unclear. In this work, we explicitly narrow the distribution of text representations by aligning them with the same prior distribution. We theoretically prove that narrowing the distribution of text representations in asymmetrical domains text matching is equivalent to optimizing the information bottleneck (IB). Since the interaction between text representations plays an important role in asymmetrical domains text matching, IB does not restrict the interaction between text representations. Therefore, we propose the adequacy of interaction and the incompleteness of a single text representation on the basis of IB and obtain the representation matching information bottleneck (RMIB). We theoretically prove that the constraints on text representations in RMIB is equivalent to maximizing the mutual information between text representations on the premise that the task information is given. On four text matching models and five text matching datasets, we verify that RMIB can improve the performance of asymmetrical domains text matching.

## 1 INTRODUCTION

Text matching is a basic task in natural language processing (NLP). The purpose of this task is to predict the semantic relationship between two input texts. Text matching can be applied to many natural language processing scenarios. For example, in question-and-answer matching, it is necessary to judge whether the given candidate answers match the given question (Tan et al., 2016); In natural language reasoning, for a given hypothesis and premise, it is necessary to judge whether the hypothesis and premise are entailment, contradiction or neutral (Bowman et al., 2015); In information retrieval, it is necessary to determine the correlation between the searched information and the returned candidate (Reimers & Gurevych, 2019).

Asymmetrical text matching refers to two input texts from different domains. For example, the question and the candidate answers can be viewed as being sampled from two different distributions in question-and-answer matching. One approach of asymmetrical domains text matching is to process text from different domains into text representation through deep learning models. In the common semantic space, text representations from different domains have the same dimensions, so it is easy to define semantic matching functions. The current mainstream approaches mainly focus on how to design more effective models to increase the interaction between input texts from different domains to obtain better text representations (Chen et al., 2017; Wang et al., 2017; Gong et al.; Kim et al., 2019; Yang et al., 2019). In the actual text matching, (Yu et al., 2022) finds an interesting phenomenon that the text representations from different domains are usually separated in the early stage of training. However, with the deepening of training, the text representations from different domains gradually begin to mix with each other. The explanation for this phenomenon is that there is a lexical gap between texts from different domains (Tay et al., 2018), so it is necessary to bridge this

gap in the process of learning, which is similar to cross-modal matching. Existing researches (Li et al., 2020; Hazarika et al., 2020) shows that the matching model should generate domain invariant features, that is, the global distribution of text representations from different domains in the common semantic space should be as similar as possible, so as to capture the commonality between potential text representations. However, existing text matching models lack explicit constraints to make text representations domain invariant. It is still an open problem to make it clear how to align the text representation distribution of different domains in the common semantic space. Meanwhile, the role of aligning the distribution of text representation from different domains remains unclear. (Yu et al., 2022) designs a regularizer based on distributed distance to explicitly promote domain invariant features. Unlike (Yu et al., 2022), which directly optimizes the distribution of text representations from different domains, we reduce the distance between the distribution of text representations from different domains by explicitly aligning the distribution of text representations from different domains with the same prior distribution. We theoretically prove that narrowing the distribution of text representations in asymmetrical domains text matching is equivalent to optimizing the information bottleneck (IB). For text matching, the interaction between texts has an important impact on the matching effect (Chen et al., 2017; Wang et al., 2017; Gong et al.; Kim et al., 2019; Yang et al., 2019). However, IB does not restrict the interaction between text representations. Therefore, we propose the adequacy of interaction and the incompleteness of a single text representation on the basis of IB and obtain the representation matching information bottleneck (RMIB). We theoretically prove that the constraints on text representations in RMIB is equivalent to maximizing the mutual information between text representations on the premise that the task information is given. The contributions of this work can be summarized as follows:

- We improve domain invariant features by explicitly aligning the text representations distribution from different domains with the same prior distribution. We theoretically prove that narrowing the text representations distribution is equivalent to optimizing the bottleneck information (IB) in asymmetrical domains text matching.

- We propose the adequacy of interaction and the incompleteness of a single text representation for text representations on the basis of IB and obtain the representation matching information bottleneck (RMIB).

- We theoretically prove that the constraints on text representations in RMIB is equivalent to maximizing the mutual information between text representations on the premise that the task information is given.

- We give the concrete implementation method of optimizing RMIB. On four text matching models and five text matching datasets, we verify that RMIB can improve the performance of asymmetrical domains text matching.

## 2 METHODOLOGY

### 2.1 NOTATION AND DEFINITIONS

We use capital letters for random variables and lower-case letters for realizations. Let $X$ and $Y$ be random variables, then the information entropy of random variable $X$ is defined as $H(X) = \mathbb{E}_{p(x)}[-\log p(x)]$. The definition of conditional entropy is $H(Y|X) = \mathbb{E}_{p(x,y)}[-\log p(y|x)]$. The mutual information between $X$ and $Y$ is defined as $I(X;Y) = H(X) - H(X|Y)$. The Kullback-Leibler (KL) divergence between two distributions is defined as $D_{\mathrm{KL}}(p(x) \parallel q(y)) = \mathbb{E}_{p(x)}[\log \frac{p(x)}{q(y)}]$.

### 2.2 INFORMATION BOTTLENECK

Information bottleneck (IB) (Tishby et al., 2000), also known as representation information bottleneck (RIB), regards the feedforward calculation process of deep neural network as a Markov chain. Let $X$ be the model input, $Y$ is the task label, $\hat{Y}$ is the prediction result of the model and $Z$ is the output of the middle layer, that is, $Z$ is the representation of the model input. If the above variables are regarded as random variables, the following Markov chain is formed:

$$Y \to X \to Z \to \hat{Y} \tag{1}$$

This means that $Y \perp Z \mid X$ or $P(Z \mid X) = P(Z \mid X, Y)$ is satisfied, that is, under the premise that $X$ is known, the existence of $Y$ will not affect $Z$, indicating that $Y$ and $Z$ are independent of each other. IB put forward the following two requirements for the representation Z learned by the deep learning model.

**Sufficient**: The representation $Z$ should contain as much information related to the target label as possible.

$$I(Y; Z) = I(X; Y) \tag{2}$$

**Minimal**: The representation $Z$ should contain as little information as possible about input $X$.

$$\min I(X; Z) \tag{3}$$

On the premise of meeting the sufficient, IB requires that the representation $Z$ should contain as little information as possible in input $X$, which means that it contains as little redundant information as possible. Combining sufficient and minimal, we can get the optimization objective of IB as follows:

$$Z^* = \arg \min_Z I(X; Z) \ s.t. \ I(Y; Z) = I(X; Y) \tag{4}$$

By using the Lagrange multiplier method, we can remove the constraint and equivalently express the above formula as:

$$Z^* = \arg \min_Z I(X; Z) + \beta(\underbrace{I(X; Y)}_{contant} - I(Y; Z))$$

$$\propto \arg \min_Z I(X; Z) - \beta I(Y; Z) \tag{5}$$

where $\beta$ is a positive constant. The optimization objective of Equation (5) is also called IB Lagrangian.

### 2.3 Information Bottleneck in Text Matching

In the asymmetrical domains text matching task, the input of the model contains two texts, which are represented by two random variables, $X_1$ and $X_2$. Usually $X_1$ and $X_2$ are sampled from different domains. Then the asymmetrical domains text matching task can be expressed as:

$$\theta^* = \arg \max_\theta \mathbb{E}_{p(x_1, x_2, y)}[\log p_\theta(y \mid x_1, x_2)] \tag{6}$$

where $\theta$ represents the parameters of the constructed text matching model, $x_1 \in X_1$, $x_2 \in X_2$ and $y \in Y$. The samples are assumed to come from an unknown distribution $p(x_1, x_2, y)$. The current text matching mainly includes two types of matching models: text encoding model and text interaction model. The text encoding model will encode the input texts $X_1$ and $X_2$ respectively, and there will be no interaction between $X_1$ and $X_2$ during the encoding process. A typical model of text encoding model is siamese neural network (Koch et al., 2015). The text interaction model will interact with the text representation in the feedforward calculation, and usually get better results. A typical text interaction model is ESIM (Chen et al., 2017). However, no matter which text matching model is used, two input text representations will be obtained in the feedforward calculation process. Therefore, the Markov chain in the process of text matching can be expressed as:

$$Y \to (X_1, X_2) \to (Z_1, Z_2) \to Z \to \hat{Y} \tag{7}$$

where $Z_1$ and $Z_2$ are the representation of $X_1$ and $X_2$ respectively and $Z$ represents the final representation after the fusion of two text representations, usually the output of the representation fusion layer. Accordingly, the optimization objective of the information bottleneck in the text matching can be expressed as:

$$Z_1^*, Z_2^* = \arg \min_{Z_1, Z_2} I(X_1, X_2; Z_1) + I(X_1, X_2; Z_2) \ s.t. \ I(Y; Z_1, Z_2) = I(Y; X) \tag{8}$$

### 2.4 Text Domain Alignment Based on Prior Distribution

Recent studies have shown that domain matching of text representation will help improve the generalization ability of text matching (Yu et al., 2022). To effectively align the distribution of text representation, (Yu et al., 2022) designs a regularizer based on distributed distance to narrow the text representations:

$$\theta^* = \arg \min_\theta \mathbb{E}_{p(x_1, x_2, y)}[-\log p_\theta(y \mid x_1, x_2)] \ s.t. \ D(p_\theta(z_1 \mid x_1, x_2), p_\theta(z_2 \mid x_1, x_2)) = 0 \tag{9}$$

where $D(\cdot)$ is a metric function of distribution distance. Unlike (Yu et al., 2022), we narrow the distribution between text representations by explicitly aligning text representations with a prior distribution in text matching. Specifically, let the prior distribution be $p(\mathcal{Z})$ and two input text representations be $Z_1$ and $Z_2$. To make the distribution between $Z_1$ and $Z_2$ as close as possible, we expect the distribution of $Z_1$ and $Z_2$ to be as close as $p(\mathcal{Z})$ on the premise that the model can correctly predict the target label. Therefore, the optimization objective of text matching based on text representations alignment can be expressed as:

$$\theta^* = \arg\min_\theta \mathbb{E}_{p(x_1,x_2,y)}[-\log p_\theta(y \mid x_1, x_2)] \; s.t. \; \begin{cases} D_{\mathrm{KL}}(p_\theta(z_1 \mid x_1, x_2) \parallel p(\mathcal{Z})) = 0 \\ D_{\mathrm{KL}}(p_\theta(z_2 \mid x_1, x_2) \parallel p(\mathcal{Z})) = 0 \end{cases} \quad (10)$$

We can use our method to explain DDR-Match (Yu et al., 2022). Let $p(\hat{z})$ be the distribution after DDR-Match alignment, then in our method it can be expressed as:

$$\theta^* = \arg\min_\theta \mathbb{E}_{p(x_1,x_2,y)}[-\log p_\theta(y \mid x_1, x_2)] \; s.t. \; \begin{cases} D_{\mathrm{KL}}(p_\theta(z_1 \mid x_1, x_2) \parallel p(\hat{z})) = 0 \\ D_{\mathrm{KL}}(p_\theta(z_2 \mid x_1, x_2) \parallel p(\hat{z})) = 0 \end{cases} \quad (11)$$

By using the Lagrange multiplier method, we can remove the constraint and equivalently express the above formula as:

$$\theta^* = \arg\min_\theta \mathbb{E}_{p(x_1,x_2,y)}[\beta_1 D_{\mathrm{KL}}(p_\theta(z_1 \mid x_1, x_2) \parallel p(\mathcal{Z})) + \beta_2 D_{\mathrm{KL}}(p_\theta(z_2 \mid x_1, x_2) \parallel p(\mathcal{Z}))$$
$$- \log p_\theta(y \mid x_1, x_2)] \quad (12)$$

where $\beta_1$ and $\beta_2$ are positive constants. Although some studies have shown that domain alignment of text representations will help improve the performance of text matching, the role of text representations distribution alignment is still unclear. We analyze this phenomenon and get Proposition 1. The proof of Proposition 1 can be found in appendix. Proposition 1 shows that domain matching in text matching is equivalent to optimizing the information bottleneck in text matching, which indicates that domain alignment of input texts can make the learned text representation forget the input redundant information as much as possible.

**Proposition 1.** *Given matching texts $X_1, X_2$, text representations $Z_1, Z_2$ and task label $Y$, then it satisfies in asymmetrical domains text matching:*

$$Z_1^*, Z_2^* \propto \arg\min_{Z_1,Z_2} I(X_1, X_2; Z_1) + I(X_1, X_2; Z_2) - \beta I(Y; Z_1, Z_2)$$

$$\propto \arg\min_{z_1,z_2} \mathbb{E}_{p(x_1,x_2,y)}[\beta_1 D_{\mathrm{KL}}(p_\theta(z_1 \mid x_1, x_2) \parallel p(\mathcal{Z})) + \beta_2 D_{\mathrm{KL}}(p_\theta(z_2 \mid x_1, x_2) \parallel p(\mathcal{Z})) - \log p_\theta(y \mid x_1, x_2)]$$

*where $\beta_1, \beta_2$ and $\beta$ are positive constants.*

## 2.5 Representation Matching Information Bottleneck

Information bottleneck (IB) only requires that the learned representations retain as much information as possible about the task label and as little information as possible about the input and does not impose any explicit constraints on the interaction between text representations. In text matching, the performance of text interaction model is usually better than that of text encoding model, which indicates that interaction on text representations has an important impact on the result of text matching. Therefore, we extend IB and obtain representation matching information bottleneck (RMIB) in text matching. Specifically, RMIB's constraints on text representations and input are consistent with IB, but RMIB puts forward the following constraints on the interaction between text representations and the relationship between text representations and task label.

**Sufficient**: The representations $Z_1$ and $Z_2$ should contain as much information related to the target label as possible. This constraint is consistent with the Sufficient of IB.

$$I(Y; Z_1, Z_2) = I(X; Y) \quad (13)$$

**Interaction**: The interaction between text representations should be sufficient, which means there should be enough mutual information between the two text representations.

$$\max I(Z_1; Z_2) \quad (14)$$

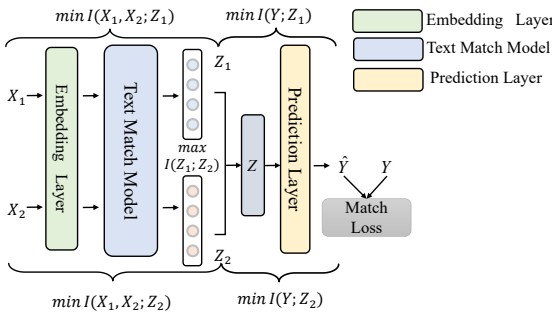

Figure 1: The optimization objective of RMIB. $\min I(X_1, X_2; Z_1)$ and $\min I(X_1, X_2; Z_2)$ indicate that text representations $Z_1$ and $Z_2$ should contain as little input information as possible. $\min I(Y; Z_1)$ and $\min I(Y; Z_2)$ indicate that a single text representation cannot complete correct text matching. $\max I(Z_1; Z_2)$ explicitly requires as much interaction as possible between $Z_1$ and $Z_2$.

**Inadequacy**: The final correct result cannot be obtained only by using a single text representation in text matching. In text matching, the model needs to make right decisions based on text representation of two input texts, which means that only using a single text representation can not get the right result. For example, in the text encoding model, due to the lack of interaction between the two input texts during the feedforward calculation, it is extremely absurd to use only a single text representation to accurately predict the target.

$$\min I(Y; Z_1) + I(Y; Z_2) \tag{15}$$

Based on the constraints proposed by RMIB for text representations, we can obtain the optimization objective of RMIB:

$$Z_1^*, Z_2^* = \arg \min_{Z_1, Z_2} I(X_1, X_2; Z_1) + I(X_1, X_2; Z_2)$$
$$s.t. \ \max I(Z_1; Z_2) + I(Z_1, Z_2; Y) - (I(Y; Z_1) + I(Y; Z_2)) \tag{16}$$

A more intuitive RMIB optimization objective is shown in Figure 1. Although RMIB requires the interaction between text representations should be sufficient, it is difficult to know the impact of the interaction of text representations on task label if we only consider the interaction of text representations without associating with task label. Further, we prove that the constraints of RMIB on text representation and task is equivalent to maximizing the mutual information between text representations on the premise that the task information is given and get the Proposition 2. The proof of Proposition 2 can be found in appendix. Proposition 2 shows that the optimization objective of RMIB can be expressed as:

$$Z_1^*, Z_2^* = \arg \min_{Z_1, Z_2} I(X_1, X_2; Z_1) + I(X_1, X_2; Z_2) \ s.t. \ \max I(Z_1; Z_2 \mid Y) \tag{17}$$

Although we can see the differences between RMIB and IB from the sufficiency of text representation interaction and the inadequacy of a single text representation, we can see the differences between RMIB and IB more intuitively from Proposition 2. IB only requires text representation to contain as much task information as possible, but it lacks constraints on the interaction between text representations. While RMIB requires that the interaction between text representations should be as much as possible on the premise of giving task label. This shows that RMIB is more suitable for text matching than IB. Another function of Proposition 2 is to provide a transformation method for optimizing conditional mutual information.

**Proposition 2.** *Given text representations $Z_1, Z_2$ and task label $Y$, then it satisfies in asymmetrical domain text matching:*

$$I(Z_1; Z_2|Y) = I(Z_1; Z_2) + I(Z_1, Z_2; Y) - I(Y; Z_1) - I(Y; Z_2)$$

## 2.6 EQUIVALENT OPTIMIZATION OBJECTIVE OF REPRESENTATION MATCHING INFORMATION BOTTLENECK

Although RMIB gives optimization objective in text matching based on mutual information, it is difficult to directly optimize mutual information items during training process. Fortunately, as mutual

information has received more attention in deep learning recently, some methods for approximate estimation of mutual information have also been proposed (Alemi et al., 2016; Belghazi et al., 2018; Hjelm et al.), which makes the optimization objective of RMIB tractable. For the optimization objective of RMIB, we can further equivalently express it as:

$$Z_1^*, Z_2^* = \arg \min_{Z_1, Z_2} I(X_1, X_2; Z_1) + I(X_1, X_2; Z_2) \ s.t. \ C - I(Z_1; Z_2 \mid Y) = 0 \quad (18)$$

where the constant C is the tightest upper bound of the theory of $I(Z_1; Z_2 \mid Y)$, and it can be easily proved that $C \leq \min \{ log |Z_1|, log |Z_2| \}$. The proof of Proposition 3 can be found in appendix.

**Proposition 3.** *Given text representations $Z_1, Z_2$ and task label $Y$, let $C := \sup I(Z_1; Z_2 \mid Y)$, then it satisfies in asymmetrical domains text matching:*

$$C \leq \min \{ log |Z_1|, log |Z_2| \}$$

Based on the Lagrange multiplier method and Proposition 2, we can further equivalently express Equation (18) as:

$$\begin{aligned} Z_1^*, Z_2^* = \arg \min_{Z_1, Z_2} I(X_1, X_2; Z_1) + I(X_1, X_2; Z_2) - \beta_1 I(Z_1; Z_2) - \beta_2 I(Z_1, Z_2; Y) \\ + \beta_3 (I(Y; Z_1) + I(Y; Z_2)) \end{aligned} \quad (19)$$

where $\beta_1$, $\beta_2$ and $\beta_3$ are positive constants. For $I(Z_1, Z_2; Y)$, we can also replace it with $I(Z; Y)$ where $Z$ is the fusion of $Z_1$ and $Z_2$. Proposition 4 shows that if the costs of optimizing $I(Y; Z_1, Z_2)$ and $I(Y; Z)$ to the same value are no different, then directly optimizing $I(Y; Z)$ will make the optimization target of RMIB have a smaller value, that is, a more optimized $Z_1^*$ and $Z_2^*$. The proof of Proposition 4 can be found in appendix. Based on Equation (19), Proposition 5 gives the objective function of the RMIB that is ultimately used for optimization in the training process. The proof of Proposition 5 can be found in appendix. It is worth noting that the application of RMIB is not limited to text encoding model or text interaction model, because text representations $Z_1$ and $Z_2$ can be generated in either model.

**Proposition 4.** *Given text representations $Z_1, Z_2$, fusion text representation $Z$ and task label $Y$, if the cost of optimizing $I(Y; Z_1, Z_2)$ and $I(Y; Z)$ to the same value is the same, then optimizing $I(Y; Z)$ will obtain better $Z_1^*$ and $Z_2^*$ in RMIB.*

**Proposition 5.** *The optimization objective of RMIB can be equivalently expressed as:*

$$Z_1^*, Z_2^* \propto \arg \min_{Z_1, Z_2} \mathbb{E}_{p(x_1, x_2, y)} [\alpha_1 (D_{\mathrm{KL}}(p_\theta(z_1 \mid x_1, x_2) \parallel p(\mathcal{Z})) + D_{\mathrm{KL}}(p_\theta(z_2 \mid x_1, x_2) \parallel p(\mathcal{Z})))$$

$$- \alpha_2 (\mathbb{E}_{p_\theta(z_1, z_2)}[f] - \log \mathbb{E}_{p_\theta(z_1) p_\theta(z_2)}[e^f])$$
$$- p_\theta(z_1, z_2 \mid x_1, x_2) \log p_\theta(y \mid z_1, z_2)$$
$$+ \alpha_3 (p_\theta(z_1 | x_1, x_2) \log p_\theta(y | z_1) + p_\theta(z_2 | x_1, x_2) \log p_\theta(y | z_2))]$$

*where $\alpha_1, \alpha_2$ and $\alpha_3$ are positive constants and $f$ is a multi-layer perceptron whose output layer activation function is sigmoid.*

## 3 EXPERIMENT

### 3.1 DATESET AND METRIC

**SICK** (Sentences Involving Compositional Knowledge) (Marelli et al., 2014) is a dataset for compositional distributional semantics, which consists of 9.8k pairs of sentences. Each sentence pair is labelled as either contradiction, neutral or entailment. Accuracy is used as an evaluation metric for this dataset.

**SciTail** (Science Entailment) (Khot et al., 2018) is an entailment dataset created from multiple-choice science exams and web sentences. The label could be entailment or neutral. The dataset contains a total of 27k samples, of which 10k samples have entailment labels, and the remaining 17k are marked neutral. Accuracy is used as an evaluation metric for this dataset.

**WikiQA** (Yang et al., 2015) is a retrieval-based question answering dataset based on Wikipedia. The binary label indicates whether the candidate sentence matches the question. The dataset contains

20.4k training pairs, 2.7k development pairs, and 6.2k test pairs. Mean average precision (MAP) and mean reciprocal rank (MRR) are used as the evaluation metrics for this dataset.

**SNLI** (Stanford Natural Language Inference) (Bowman et al., 2015) is a benchmark dataset for natural language inference which contains 570k human annotated sentence pairs. One of the two sentences is used as "premise" and the other is used as "hypothesis". The label is one of "entailment", "neutral", "contradiction" and "-". We follow the practice in (Bowman et al., 2015) and the data labeled "-" is deleted. Accuracy is used as an evaluation metric for this dataset.

**Quora Question Pair** is a dataset for paraphrase identification. Quora Question Pair contains 400k question pairs with binary label collected from the Quora website. This task is to determine whether two sentences are the interpretation of each other. Accuracy is used as an evaluation metric for this dataset.

## 3.2 IMPLEMENTATION DETAILS

The data preprocessing of SNLI, Quora Question Pair, SciTail and WikiQA is consistent with RE2 (Yang et al., 2019). We use 840B-300d GloVe (Pennington et al., 2014) to initialize the embedded layer of RE2 (Yang et al., 2019) and ESIM (Chen et al., 2017). We choose the standard Gaussian distribution as $p(\mathcal{Z})$, because on the premise that the mean and variance are known, the entropy of the Gaussian distribution is the maximum, that is, the minimum prior knowledge is introduced. For the KL term in RMIB, we use the reparameter trick (Kingma & Welling, 2013) to optimize it. The hyper-parameters of ESIM and RE2 is consistent with (Yang et al., 2019). For BERT (Kenton & Toutanova, 2019) and SBERT (Reimers & Gurevych, 2019), we use the max pooling to obtain the text representations, the learning rate is 2e-5 and the epoch is 6. All experiments use Adam (Kingma & Ba, 2014) as the optimizer and the hyper-parameters search range for $\alpha_1, \alpha_2$ and $\alpha_3$ in RMIB is $\{0.01, 0.02, 0.03\}$. To ensure the reproducibility of the results, all experiments use the same seed. All experiments are run on a server with 128G RAM and four RTX 2080Ti graphics cards.

## 3.3 PERFORMANCE OF RMIB

To verify the validity of RMIB, we conduct the experiment using four common text matching models on five public datasets. BERT and SBERT are pre-training models while RE2 and ESIM are non-pre-training models. Meanwhile, BERT, RE2 and ESIM are text interaction models while SBERT is text encoding model. The experimental results are shown in Table 1. From Table 1, it can be seen that compared to the vanilla model, almost all models show significant improvements on the five datasets after using RMIB. We can find that compared with BERT, RE2 and ESIM, RMIB improves SBERT significantly in the five public datasets. For example, SBERT with RMIB can achieve 12.54% gain on accuracy on SICK. Although the effect of SBERT is not as superior as that of BERT, the difference between the two decreases sharply after the addition of RMIB. For example, the F1 difference between the original SBERT and BERT on SICK is 14.64%, but after the addition of RMIB, the F1 value difference is reduced to 2.81%. Since the optimization goal of RMIB is the premise of known task information, the interaction between text representations should be as much as possible, while SBERT itself does not interact with text representations before the fusion layer. Therefore, RMIB can improve SBERT's model performance by explicitly increasing the interaction between text representations. This also indicates that for text matching tasks, the interaction between text representations is crucial to the final result.

We find that RMIB can bring greater improvement in the case of data hunger. The improvement of the four text matching models on SICK, SciTail and WikiQA is significantly greater than that on SNLI and Quora. We think this is because SNLI and Quora have significantly more data than SICK, SciTail, WikiQA. For example, SNLI has about 57 times the data than SICK. (Koltchinskii, 2001) indicates that when the number of training data sets increases, the generalization performance of the model will also increase. When there are enough training sets, the model itself has good generalization performance, so RMIB can bring less improvement. We also find that BERT's accuracy on SNLI decreased by 0.03% after adding RMIB. We believe that this may be because of the less significant heterogeneity of the text to be matched in SNLI.

Table 1: Performance of four text match model with RMIB on five public dataset. Right superscript * represents the results reproduced in our experimental environment. The other data in the table is obtained from (Yu et al., 2022). Due to the lack of results reported on Quora in (Yu et al., 2022), we use - for representation.

| Models | SICK Acc. | SciTail Acc. | SciTail F1 | WikiQA MAP | WikiQA MRR | SNLI Acc. | Quora Acc. |
|---|---|---|---|---|---|---|---|
| RE2 | 84.20 | 86.61 | 88.73 | 74.96 | 76.58 | 89.00 | - |
| RE2(DDR-Match,JS) | 84.30 | 86.74 | 88.85 | 73.93 | 76.04 | 88.75 | - |
| RE2(DDR-Match,MMD) | 84.35 | 87.39 | 89.58 | 74.84 | 76.24 | 89.02 | - |
| RE2(DDR-Match,WD) | 85.39 | 87.04 | 89.12 | 75.31 | 76.89 | 89.09 | - |
| RE2* | 84.96 | 86.27 | 88.54 | 74.28 | 76.05 | 88.94 | 89.52 |
| RE2(RMIB) | **86.53**(+1.57) | **87.68**(+1.41) | **89.82**(+1.28) | **75.54**(+1.26) | **77.73**(+1.68) | **89.19**(+0.25) | **89.85**(+0.33) |
| BERT | 86.65 | 89.56 | 91.52 | 77.52 | 78.95 | 87.91 | - |
| BERT(DDR-Match,JS) | 85.04 | 90.40 | 92.13 | 79.03 | 80.87 | 87.90 | - |
| BERT(DDR-Match,MMD) | 86.68 | 90.22 | 91.96 | 79.54 | 81,21 | 88.02 | - |
| BERT(DDR-Match,WD) | 86.72 | 90.53 | 92.29 | 79.58 | 81.23 | 88.23 | - |
| Bert* | 86.69 | 93.79 | 94.88 | 83.40 | 85.01 | **90.46** | 90.41 |
| BERT(RMIB) | **87.40**(+0.71) | **94.12**(+0.33) | **95.16**(+0.28) | **84.29**(+0.89) | **86.02**(+1.01) | 90.43(-0.03) | **90.72**(+0.31) |
| SBERT | 63.31 | 79.23 | 83.74 | 68.38 | 69.74 | 83.76 | - |
| SBERT(DDR-Match,JS) | 63.35 | 80.75 | 83.68 | 68.89 | 70.27 | 84.01 | - |
| SBERT(DDR-Match,MMD) | 65.29 | 81.67 | 84.49 | 68.85 | 70.16 | 83.91 | - |
| SBERT(DDR-Match,WD) | 64.39 | 81.93 | 84.77 | 69.04 | 70.43 | **84.17** | - |
| SBERT* | 72.05 | 83.44 | 86.27 | 71.22 | 72.67 | 80.04 | 86.46 |
| SBERT(RMIB) | **84.59**(+12.54) | **87.72**(+4.28) | **90.52**(+4.25) | **74.73**(+3.51) | **76.28**(+3.61) | 81.91(+1.87) | **87.34**(+0.88) |
| ESIM* | 82.59 | 85.04 | 87.24 | 72.38 | 74.10 | 87.51 | 88.09 |
| ESIM(RMIB) | **83.29**(+0.70) | **86.03**(+0.99) | **88.29**(+1.05) | **72.99**(+0.61) | **74.55**(+0.45) | **87.53**(+0.02) | **88.26**(+0.17) |

## 3.4 ABLATION STUDY

As shown in Proposition 5, the optimization objective of RMIB consists of four parts. In addition to cross entropy loss, we use ablation experiments on the other three items to explore the role of each item. Since RMIB is obtained by adding **Interaction** and **Inadequacy** to IB, we also perform ablation on these two items to compare the differences between RMIB and IB. By setting $\alpha_2$ and $\alpha_3$ in the RMIB optimization objective to 0, IB can be obtained. The results of the ablation experiment are shown in Table 2. From Table 2, it can be seen that in most cases, the results of ablation of the components of RMIB have decreased compared to the model with RMIB. This is consistent with Proposition 2, because the information theory interpretation of the three ablation items of RMIB is to maximize the mutual information between text representations on the premise of knowing the task information. When we ablate one of the items, the information theory interpretation is no longer valid. For RMIB and IB, we find that RMIB consistently outperforms IB on all datasets and models. This also indicates that RMIB is more suitable for text matching tasks. We find that the ablation term in RMIB plays different roles on different datasets. For example, when we set $\alpha_1$ in RMIB to 0, compared to the original model, BERT's performance on SICK decreased by 0.06%, but it improved by 0.16% on Quora. We also find that for text encoding models like SBERT, no matter how RMIB is ablated, it can always bring improvements on SICK, SciTail, and WikiQA. We believe that this is because of the small number of the three datasets and the lack of explicit interaction in SBERT itself, so increasing even some of the items in RMIB can bring gains.

## 4 RELATED WORK

Text matching can be mainly divided into two methods: text encoding model and text interaction model. Early work explored encoding each text individually as a vector and then building a neural network classifier on both vectors. In this paradigm, several different models are used to encode the input text individually. Recurrent neural networks and convolutional neural networks are used as text encoder (Bowman et al., 2015; Tai et al., 2015; Yu et al., 2014; Tan et al., 2016). SBERT (Reimers & Gurevych, 2019) uses the pre-trained model BERT (Kenton & Toutanova, 2019) as the text encoder. The text encoding model does not explicitly consider interactions between texts, while text interaction models consider adding interactions between input texts to improve performance. ESIM (Chen et al., 2017) uses attention mechanisms to interact between input texts. BiMPM (Wang et al., 2017) utilizes a variety of attention to increase the interaction between input texts. DIIN (Gong et al.) uses DenseNet (Huang et al., 2017) as a deep convolutional feature extractor to extract information from the alignment results. DRCN (Kim et al., 2019) stacks encoding layers and attention layers and then connects all previously aligned results. RE2 (Yang et al., 2019) introduces an architecture based on

Table 2: Ablation of RMIB. The superscript - indicates that we set the corresponding coefficient to 0. For example, (RMIB, $\alpha_1$, $\alpha_2^-$, $\alpha_3^-$) indicates that we set $\alpha_2$ and $\alpha_3$ to 0, in which case RMIB will degenerate to IB.

| Models | SICK Acc. | SciTail Acc. | SciTail F1 | WikiQA MAP | WikiQA MRR | SNLI Acc. | Quora Acc. |
|---|---|---|---|---|---|---|---|
| RE2 | 84.96 | 86.27 | 88.54 | 74.28 | 76.05 | 88.94 | 89.52 |
| RE2(RMIB) | **86.53**(+1.57) | **87.68**(+1.41) | **89.82**(+1.28) | **75.54**(+1.26) | **77.73**(+1.68) | **89.19**(+0.25) | 89.85(+0.33) |
| RE2(RMIB, $\alpha_1^-$, $\alpha_2$, $\alpha_3$) | 85.96(+1.00) | 86.83(+0.56) | 89.05(+0.51) | 72.01(-2.27) | 74.05(-2.05) | 86.42(-2.52) | **90.24**(+0.72) |
| RE2(RMIB, $\alpha_1$, $\alpha_2^-$, $\alpha_3$) | 86.20(+1.24) | 86.08(-0.19) | 88.77(+0.23) | 73.31(-0.97) | 75.35(-0.70) | 86.28(-2.66) | 89.67(+0.15) |
| RE2(RMIB, $\alpha_1$, $\alpha_2$, $\alpha_3^-$) | 86.16(+1.2) | 86.83(+0.56) | 89.16(+0.62) | 73.29(-0.99) | 74.88(-1.17) | 88.61(-0.33) | 89.79(+0.27) |
| RE2(RMIB, $\alpha_1$, $\alpha_2^-$, $\alpha_3^-$) | 86.02(+1.06) | 87.16(+0.89) | 89.32(+0.78) | 72.79(-1.49) | 74.42(-1.63) | 89.00(+0.06) | 89.59(+0.07) |
| BERT | 86.69 | 93.79 | 94.88 | 83.40 | 85.01 | 90.46 | 90.41 |
| BERT(RMIB) | **87.40**(+0.71) | **94.12**(+0.33) | **95.16**(+0.28) | **84.29**(+0.89) | **86.02**(+1.01) | 90.43(-0.03) | **90.72**(+0.31) |
| BERT(RMIB, $\alpha_1^-$, $\alpha_2$, $\alpha_3$) | 86.63(-0.06) | 93.18(-0.61) | 94.32(-0.56) | 83.71(+0.31) | 85.02(+0.01) | 90.27(-0.19) | 90.57(+0.16) |
| BERT(RMIB, $\alpha_1$, $\alpha_2^-$, $\alpha_3$) | 86.87(+0.18) | 93.37(-0.42) | 94.60(-0.28) | 84.56(+0.16) | 85.78(+0.77) | 90.35(-0.11) | 90.38(-0.03) |
| BERT(RMIB, $\alpha_1$, $\alpha_2$, $\alpha_3^-$) | 86.53(-0.16) | 93.56(-0.23) | 94.68(-0.20) | 83.93(+0.53) | 85.38(+0.37) | 90.31(-0.15) | 90.21(-0.20) |
| BERT(RMIB, $\alpha_1$, $\alpha_2^-$, $\alpha_3^-$) | 87.28(+0.59) | 93.60(-0.19) | 94.81(-0.07) | 83.52(+0.12) | 85.18(+0.17) | 90.03(-0.43) | 90.49(+0.08) |
| SBERT | 72.05 | 83.44 | 86.27 | 71.22 | 72.67 | 80.04 | 86.46 |
| SBERT(RMIB) | **84.59**(+12.54) | **87.72**(+4.28) | **90.52**(+4.25) | **74.73**(+3.51) | **76.28**(+3.61) | **81.91**(+1.87) | 87.34(+0.88) |
| SBERT(RMIB, $\alpha_1^-$, $\alpha_2$, $\alpha_3$) | 82.55(+10.50) | 86.92(+3.48) | 89.97(+3.70) | 72.79(+1.57) | 74.71(+2.04) | 79.78(-0.26) | **87.56**(+1.10) |
| SBERT(RMIB, $\alpha_1$, $\alpha_2^-$, $\alpha_3$) | 83.57(+11.52) | 87.39(+3.95) | 90.26(+3.99) | 73.90(+2.68) | 75.06(+2.39) | 79.42(-0.62) | 87.01(+0.55) |
| SBERT(RMIB, $\alpha_1$, $\alpha_2$, $\alpha_3^-$) | 82.59(+10.54) | 87.11(+3.67) | 90.07(+3.80) | 71.92(+0.70) | 73.72(+1.05) | 73.79(-6.26) | 86.46(+0.00) |
| SBERT(RMIB, $\alpha_1$, $\alpha_2^-$, $\alpha_3^-$) | 82.63(+10.58) | 86.69(+3.25) | 89.71(+3.44) | 73.80(+1.58) | 75.20(+2.53) | 79.64(-0.40) | 86.75(+0.29) |
| ESIM | 82.59 | 85.04 | 87.24 | 72.38 | 74.10 | 87.51 | 88.09 |
| ESIM(RMIB) | **83.29**(+0.70) | **86.03**(+0.99) | **88.29**(+1.05) | **72.99**(+0.61) | **74.55**(+0.45) | 87.53(+0.02) | **88.26**(+0.17) |
| ESIM(RMIB, $\alpha_1^-$, $\alpha_2$, $\alpha_3$) | 83.27(+0.68) | 85.65(+0.61) | 87.84(+0.60) | 71.05(-1.33) | 72.76(-1.34) | 87.12(-0.39) | 88.02(-0.07) |
| ESIM(RMIB, $\alpha_1$, $\alpha_2^-$, $\alpha_3$) | 82.98(+0.39) | 85.79(+0.75) | 87.78(+0.54) | 70.31(-2.07) | 72.36(-1.74) | **87.75**(+0.24) | 88.07(-0.02) |
| ESIM(RMIB, $\alpha_1$, $\alpha_2$, $\alpha_3^-$) | 83.04(+0.45) | 85.18(+0.14) | 87.46(+0.22) | 70.75(-1.63) | 72.03(-2.07) | 87.60(+0.09) | 87.99(-0.10) |
| ESIM(RMIB, $\alpha_1$, $\alpha_2^-$, $\alpha_3^-$) | **83.29**(+0.70) | 84.67(-0.37) | 86.76(-0.48) | 70.53(-1.85) | 72.76(-1.34) | 87.49(-0.02) | 88.08(-0.01) |

convolutional layer and attention layer reinforcement residual connection. DDR-Match (Yu et al., 2022) is a regularizer based on distribution distance to directly narrow the encoding distribution of input texts.

The information bottleneck (IB) is proposed by (Tishby et al., 2000) as a generalization of the minimum adequacy statistic, allowing trade-offs between the adequacy and complexity of the representation. Later (Tishby & Zaslavsky, 2015) and (Shwartz-Ziv & Tishby, 2017) advocate using IB between test data and deep neural network activation to study the adequacy and minimity of the representation. Since it is difficult to directly calculate mutual information, VIB (Alemi et al., 2016) gives the lower bound of the optimization goal for the information bottleneck based on the variational method, so that the optimization goal of the information bottleneck can be combined into the training process. DisenIB (Pan et al., 2021) introduces disentangled information bottleneck from the perspective of supervised detangling. PIB (Wang et al., 2021) establishes an information bottleneck between the accuracy of neural networks and the complexity of information based on information stored in weights.

## 5   Conclusion and Limitations

In this paper, we theoretically prove that narrowing text representations distribution in asymmetrical domains text matching is equivalent to optimizing the information bottleneck. Since the information bottleneck does not constrain the interaction of text representations, we extend the information bottleneck in asymmetrical domain text matching and obtain the representation matching information bottleneck (RMIB). We also theoretically prove that the optimization objective of RMIB is equivalent to maximizing mutual information between text representations given task information. The experimental results show the effectiveness of RMIB.

We verify the validity of RMIB in asymmetrical domain text matching, but its effectiveness in some heterogeneous data matching scenarios is not verified, where the inconsistency of data distribution to be matched increases significantly. We will explore the effectiveness of RMIB in heterogeneous data matching scenarios in future work. Another limitation is that there are some hyper-parameters in the optimization goal of RMIB that need to be set manually. We will also explore how to set the hyper-parameters adaptively in future work.

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

APPENDIX

PROOF OF PROPOSITION 1

**Proposition 1.** *Given matching texts $X_1, X_2$, text representations $Z_1, Z_2$ and task label $Y$, then it satisfies in asymmetrical domains text matching:*

$$Z_1^*, Z_2^* \propto \arg \min_{Z_1, Z_2} I(X_1, X_2; Z_1) + I(X_1, X_2; Z_2) - \beta I(Y; Z_1, Z_2)$$

$$\propto \arg \min_{z_1, z_2} \mathbb{E}_{p(x_1, x_2, y)}[\beta_1 D_{\mathrm{KL}}(p_\theta(z_1 \mid x_1, x_2) \parallel p(\mathcal{Z})) + \beta_2 D_{\mathrm{KL}}(p_\theta(z_2 \mid x_1, x_2) \parallel p(\mathcal{Z})) - \log p_\theta(y \mid x_1, x_2)]$$

*where $\beta_1, \beta_2$ and $\beta$ are positive constants.*

*Proof.* The optimization objective of the representation information bottleneck in the text matching can be expressed as:

$$Z_1^*, Z_2^* = \arg \min_{Z_1, Z_2} I(X_1, X_2; Z_1) + I(X_1, X_2; Z_2) \ s.t. \ I(Y; Z_1, Z_2) = I(Y; X) \tag{20}$$

$$\propto \arg \min_{Z_1, Z_2} I(X_1, X_2; Z_1) + I(X_1, X_2; Z_2) - \beta I(Y; Z_1, Z_2) \tag{21}$$

For $I(Z_1; X_1, X_2)$, we can get:

$$I(Z_1; X_1, X_2) = \sum_{x_1, x_2, z_1} p_\theta(x_1, x_2, z_1) \log \frac{p_\theta(z_1 | x_1, x_2)}{p_\theta(z_1)} \tag{22}$$

$$= \sum_{x_1, x_2, z_1} p_\theta(x_1, x_2, z_1) \log p_\theta(z_1 | x_1, x_2) - \sum_{x_1, x_2, z_1} p_\theta(x_1, x_2, z_1) \log p_\theta(z_1) \tag{23}$$

We can replace $p_\theta(z_1)$ by the prior distribution $p(\mathcal{Z})$ and according to Gibbs inequality, we can get:

$$- \sum_{x_1, x_2, z_1} p_\theta(x_1, x_2, z_1) \log p_\theta(z_1) \leq - \sum_{x_1, x_2, \mathcal{Z}} p_\theta(x_1, x_2, z_1) \log p(\mathcal{Z}) \tag{24}$$

Further, we can get:

$$I(Z_1; X_1, X_2) \leq \sum_{x_1, x_2, z_1} p_\theta(x_1, x_2, z_1) \log p_\theta(z_1 | x_1, x_2) - \sum_{x_1, x_2, \mathcal{Z}} p_\theta(x_1, x_2, z_1) \log p(\mathcal{Z}) \tag{25}$$

$$= \sum_{x_1, x_2, z_1, \mathcal{Z}} p_\theta(x_1, x_2, z_1) \log \frac{p_\theta(z_1 | x_1, x_2)}{p(\mathcal{Z})} \tag{26}$$

$$= \sum_{x_1, x_2, z_1, \mathcal{Z}} p_\theta(x_1, x_2) p_\theta(z_1 | x_1, x_2) \log \frac{p_\theta(z_1 | x_1, x_2)}{p(\mathcal{Z})} \tag{27}$$

$$= \mathbb{E}_{p(x_1, x_2)}[D_{\mathrm{KL}}(p_\theta(z_1 | x_1, x_2) \parallel p(\mathcal{Z}))] \tag{28}$$

We can see that when $\mathbb{E}_{p(x_1, x_2)}[D_{\mathrm{KL}}(p_\theta(z_1 | x_1, x_2) \parallel p(\mathcal{Z}))]$ is minimized, $I(Z_1; X_1, X_2)$ will also be minimized.

$$\min \mathbb{E}_{p(x_1, x_2)}[\beta_1 D_{\mathrm{KL}}(p_\theta(z_1 | x_1, x_2) \parallel p(\mathcal{Z}))] \propto \min I(Z_1; X_1, X_2) \tag{29}$$

Based on the same proof method, we can also get:

$$\min \mathbb{E}_{p(x_1, x_2)}[\beta_2 D_{\mathrm{KL}}(p_\theta(z_2 | x_1, x_2) \parallel p(\mathcal{Z}))] \propto \min I(Z_2; X_1, X_2) \tag{30}$$

For $I(Z_1, Z_2; Y)$, we can get:

$$I(Z_1, Z_2; Y) = \sum_{z_1, z_2, y} p_\theta(z_1, z_2, y) \log \frac{p_\theta(y|z_1, z_2)}{p_\theta(y)} \tag{31}$$

$$= \sum_{z_1, z_2, y} p_\theta(z_1, z_2, y) \log p_\theta(y|z_1, z_2) - \underbrace{\sum_{z_1, z_2, y} p_\theta(z_1, z_2, y) \log p_\theta(y)}_{H(Y) \geq 0} \tag{32}$$

$$\geq \sum_{z_1, z_2, y} p_\theta(z_1, z_2, y) \log p_\theta(y|z_1, z_2) \tag{33}$$

Since $z_1$ and $z_2$ is the representation of $x_1$ and $x_2$, we can get:

$$p_\theta(z_1, z_2, y) = p_\theta(x_1, x_2, y) \tag{34}$$
$$p_\theta(y|z_1, z_2) = p_\theta(y|x_1, x_2) \tag{35}$$

Further, we can get:

$$I(Z_1, Z_2; Y) \geq \sum_{z_1, z_2, y, x_1, x_2} p_\theta(x_1, x_2, y) \log p_\theta(y|x_1, x_2) \tag{36}$$

$$= \mathbb{E}_{p(y, x_1, x_2)}[\log p_\theta(y|z_1, z_2)] \tag{37}$$

We can find that when maximizing $I(Z_1, Z_2; Y)$, $\mathbb{E}_{p(y, x_1, x_2)}[\log p_\theta(y|z_1, z_2)]$ will also be maximized.

$$\max \beta I(Z_1, Z_2; Y) \propto \max \mathbb{E}_{p(y, x_1, x_2)}[\log p_\theta(y|z_1, z_2)] \tag{38}$$

Then we can get:

$$Z_1^*, Z_2^* \propto \min_{Z_1, Z_2} I(X_1, X_2; Z_1) + I(X_1, X_2; Z_2) - \beta I(Y; Z_1, Z_2) \tag{39}$$

$$\propto \min_{z_1, z_2} \mathbb{E}_{p(x_1, x_2, y)}[\beta_1 D_{\mathrm{KL}}(p_\theta(z_1 \mid x_1, x_2) \parallel p(\mathcal{Z})) + \beta_2 D_{\mathrm{KL}}(p_\theta(z_2 \mid x_1, x_2) \parallel p(\mathcal{Z})) - \log p_\theta(y \mid x_1, x_2)] \tag{40}$$

Therefor, the Proposition 1 can be proved. $\qquad \square$

PROOF OF PROPOSITION 2

**Proposition 2.** *Given text representations $Z_1, Z_2$ and task label $Y$, then it satisfies in asymmetrical domain text matching:*

$$I(Z_1; Z_2|Y) = I(Z_1; Z_2) + I(Z_1, Z_2; Y) - I(Y; Z_1) - I(Y; Z_2)$$

*Proof.* According to the definition of conditional mutual information, we can get:

$$I(Z_1; Z_2|Y) = H(Z_1|Y) - H(Z_1|Y, Z_2) \tag{41}$$

We can decompose $H(Z_1|Y)$ and $H(Z_1|Y, Z_2)$ into:

$$H(Z_1|Y) = H(Z_1, Y) - H(Y) \tag{42}$$
$$H(Z_1|Y, Z_2) = H(Y, Z_1, Z_2) - H(Y, Z_2) \tag{43}$$

So we can get:

$$I(Z_1; Z_2|Y) = H(Z_1, Y) - H(Y) - H(Y, Z_1, Z_2) + H(Z_2, Y) \tag{44}$$

We decompose $H(Z_1, Y), H(Y, Z_1, Z_2)$ and $H(Z_2, Y)$ into:

$$H(Z_1, Y) = H(Z_1) + H(Y|Z_1) \tag{45}$$
$$H(Y, Z_1, Z_2) = H(Z_1, Z_2) + H(Y|Z_1, Z_2) \tag{46}$$
$$H(Z_2, Y) = H(Z_2) + H(Y|Z_2) \tag{47}$$

Further, we can get:

$$I(Z_1; Z_2|Y) = H(Z_1) + H(Y|Z_1) - H(Y) - H(Z_1, Z_2) - H(Y|Z_1, Z_2) + H(Z_2) + H(Y|Z_2) \tag{48}$$

We note the following relationship between $H(Z_1), H(Z_2)$ and $H(Z_1, Z_2)$:

$$H(Z_1) + H(Z_2) - H(Z_1, Z_2) = I(Z_1, Z_2) \tag{49}$$

$I(Z_1, Z_2|Y)$ can be expressed as:

$$I(Z_1; Z_2|Y) = I(Z_1, Z_2) + H(Y|Z_1) - H(Y) - H(Y|Z_1, Z_2) + H(Y|Z_2) \tag{50}$$

We further decompose $H(Y|Z_1), H(Y|Z_1, Z_2)$ and $H(Y|Z_2)$ into:

$$H(Y|Z_1) = H(Y) - I(Z_1; Y) \tag{51}$$
$$H(Y|Z_1, Z_2) = H(Y) - I(Y; Z_1, Z_2) \tag{52}$$
$$H(Y|Z_2) = H(Y) - I(Z_2; Y) \tag{53}$$

Finally, we can get:

$$I(Z_1; Z_2|Y) = I(Z_1; Z_2) + I(Z_1, Z_2; Y) - I(Y; Z_1) - I(Y; Z_2) \tag{54}$$

Therefor, the Proposition 2 can be proved. $\qquad\square$

PROOF OF PROPOSITION 3

**Proposition 3.** *Given text representations $Z_1, Z_2$ and task label $Y$, let $C := \sup I(Z_1; Z_2 \mid Y)$, then it satisfies in asymmetrical domains text matching:*

$$C \leq \min \{log |Z_1|, log |Z_2|\}$$

*Proof.* According to the definition of conditional mutual information, we can get:

$$I(Z_1; Z_2|Y) = H(Z_1|Y) - \underbrace{H(Z_1|Y, Z_2)}_{\geq 0} \tag{55}$$

$$\leq H(Z_1|Y) \tag{56}$$

Since $I(Z_1; Y) = H(Z_1) - H(Z_1|Y) \geq 0$, we can get:

$$I(Z_1; Z_2|Y) \leq H(Z_1) \tag{57}$$

Let distribution $q(Z_1) = \frac{1}{|z_1|}$, then:

$$D_{\mathrm{KL}}(p_\theta(Z_1) \parallel q(Z_1)) = \sum_{z_1} p_\theta(z_1) \log \frac{p_\theta(z_1)}{q(z_1)} \tag{58}$$

$$= \sum_{z_1} p_\theta(z_1) \log p_\theta(z_1) - \sum_{z_1} p_\theta(z_1) \log q(z_1) \tag{59}$$

$$= -\sum_{z_1} p_\theta(z_1) \log q(z_1) - \underbrace{(-\sum_{z_1} p_\theta(z_1) \log p_\theta(z_1))}_{H(Z_1)} \tag{60}$$

$$= -\sum_{z_1} p_\theta(z_1) \log \frac{1}{|Z_1|} - H(Z_1) \tag{61}$$

$$= \log |Z_1| - H(Z_1) \geq 0 \tag{62}$$

where $|z_1|$ is the number of elements of random variable $Z_1$. Therefore, we can get:

$$I(Z_1; Z_2|Y) \leq \log |Z_1| \tag{63}$$

Since mutual information meets the exchangeability, it can be obtained:

$$I(Z_1; Z_2|Y) \leq \log |Z_2| \tag{64}$$

Finally, we can get:

$$I(Z_1; Z_2|Y) \leq \min \{log |Z_1|, log |Z_2|\} \tag{65}$$

Therefor, the Proposition 3 can be proved.

$\square$

PROOF OF PROPOSITION 4

**Proposition 4.** *Given text representations $Z_1, Z_2$, fusion text representation $Z$ and task label $Y$, if the cost of optimizing $I(Y; Z_1, Z_2)$ and $I(Y; Z)$ to the same value is the same, then optimizing $I(Y; Z)$ will obtain better $Z_1^*$ and $Z_2^*$ in RMIB.*

*Proof.* According to the definition of mutual information:

$$I(Y; Z_1, Z_2, Z) = I(Y; Z_1, Z_2) + I(Y; Z|Z_1, Z_2) \tag{66}$$
$$= I(Y; Z) + I(Y; Z_1, Z_2|Z) \tag{67}$$

Since $Y \rightarrow (X_1, X_2) \rightarrow (Z_1, Z_2) \rightarrow Z$ forms a Markov chain, $Y$ and $Z$ are independent of each other given $(Z_1, Z_2)$, that is, $I(Y; Z|Z_1, Z_2) = 0$, then:

$$I(Y; Z_1, Z_2, Z) = I(Y; Z_1, Z_2) + \underbrace{I(Y; Z|Z_1, Z_2)}_{0} = I(Y; Z) + \underbrace{I(Y; Z_1, Z_2|Z)}_{\geq 0} \tag{68}$$

Based on the non-negativity of mutual information, then:

$$I(Y; Z_1, Z_2) \geq I(Y; Z) \tag{69}$$

We can see that $I(Y; Z_1, Z_2)$ is the upper bound of $I(Y; Z)$. Therefore, if the cost of optimizing $I(Y; Z_1, Z_2)$ and $I(Y; Z)$ to the same value is the same, this means that optimizing $I(Y; Z)$ will result in a higher $I(Y; Z_1, Z_2)$. Since one of the optimization goals of RMIB is to maximize $I(Y; Z_1, Z_2)$, the higher the value of $I(Y; Z_1, Z_2)$, the better $Z_1^*, Z_2^*$ learned.

$\square$

PROOF OF PROPOSITION 5

**Proposition 5.** *The optimization objective of RMIB can be equivalently expressed as:*

$$Z_1^*, Z_2^* \propto \arg \min_{Z_1, Z_2} \mathbb{E}_{p(x_1, x_2, y)}[\alpha_1(D_{\mathrm{KL}}(p_\theta(z_1 \mid x_1, x_2) \parallel p(\mathcal{Z})) + D_{\mathrm{KL}}(p_\theta(z_2 \mid x_1, x_2) \parallel p(\mathcal{Z})))$$

$$- \alpha_2(\mathbb{E}_{p_\theta(z_1, z_2)}[f] - \log \mathbb{E}_{p_\theta(z_1)p_\theta(z_2)}[e^f])$$
$$- p_\theta(z_1, z_2 \mid x_1, x_2) \log p_\theta(y \mid z_1, z_2)$$
$$+ \alpha_3(p_\theta(z_1|x_1, x_2) \log p_\theta(y|z_1) + p_\theta(z_2|x_1, x_2) \log p_\theta(y|z_2))]$$

*where $\alpha_1, \alpha_2$ and $\alpha_3$ are positive constants and $f$ is a multi-layer perceptron whose output layer activation function is sigmoid.*

*Proof.* The optimization objective of RMIB is:

$$Z_1^*, Z_2^* = \arg \min_{Z_1, Z_2} I(X_1, X_2; Z_1) + I(X_1, X_2; Z_2) - \beta_1 I(Z_1; Z_2) - \beta_2 I(Z_1, Z_2; Y) + \beta_3 (I(Y; Z_1) + I(Y; Z_2)) \tag{70}$$

$$\propto \arg \min_{Z_1, Z_2} \alpha_1 (I(X_1, X_2; Z_1) + I(X_1, X_2; Z_2)) - \alpha_2 I(Z_1; Z_2) - I(Z_1, Z_2; Y) + \alpha_3 (I(Y; Z_1) + I(Y; Z_2)) \tag{71}$$

$H(Y)$ is a constant when given a dataset. From Proposition 1 we can see:

$$\min \begin{cases} I(X_1, X_2; Z_1) \\ I(X_1, X_2; Z_2) \end{cases} \propto \min \begin{cases} D_{\mathrm{KL}}(p_\theta(z_1 \mid x_1, x_2) \parallel p(\mathcal{Z})) \\ D_{\mathrm{KL}}(p_\theta(z_2 \mid x_1, x_2) \parallel p(\mathcal{Z})) \end{cases} \tag{72}$$

$$\max I(Z_1, Z_2; Y) \propto \max \mathbb{E}_{x_1, x_2, y}[p_\theta(z_1, z_2 \mid x_1, x_2) \log p_\theta(y \mid z_1, z_2)] \tag{73}$$

$$\min \begin{cases} I(Y; Z_1) \\ I(Y; Z_2) \end{cases} \propto \min \begin{cases} \mathbb{E}_{x_1, x_2, y}(p_\theta(z_1 \mid x_1, x_2) \log p_\theta(y \mid z_1)) \\ \mathbb{E}_{x_1, x_2, y}(p_\theta(z_2 \mid x_1, x_2) \log p_\theta(y \mid z_2)) \end{cases} \tag{74}$$

According to the definition of mutual information and KL divergence, for $I(Z_1; Z_2)$ there is:

$$I(Z_1; Z_2) = D_{\mathrm{KL}}(p_\theta(z_1, z_2) \parallel p_\theta(z_1)p_\theta(z_2)) \tag{75}$$

Let $\tilde{p}(z_1, z_2)$ be defined as:

$$\tilde{p}(z_1, z_2) := \frac{1}{\phi} e^f p_\theta(z_1) p_\theta(z_2) \tag{76}$$

where $\phi := \mathbb{E}_{p_\theta(z_1)p_\theta(z_2)}[e^f]$ and $f$ is a given function. By construction:

$$\mathbb{E}_{p_\theta(z_1, z_2)}[f] - \log \phi = \mathbb{E}_{p_\theta(z_1, z_2)}[\log e^f - \log \phi] \tag{77}$$

$$= \mathbb{E}_{p_\theta(z_1, z_2)}[\log \frac{e^f}{\phi}] \tag{78}$$

$$= \mathbb{E}_{p_\theta(z_1, z_2)}[\log \frac{\tilde{p}(z_1, z_2)}{p_\theta(z_1)p_\theta(z_2)}] \tag{79}$$

We can note that:

$$D_{\mathrm{KL}}(p_\theta(z_1, z_2) \parallel p_\theta(z_1)p_\theta(z_2)) - (\mathbb{E}_{p_\theta(z_1, z_2)}[f] - \log \phi) = \mathbb{E}_{p_\theta(z_1, z_2)}[\log \frac{p_\theta(z_1, z_2)}{p_\theta(z_1)p_\theta(z_2)}] - \mathbb{E}_{p_\theta(z_1, z_2)}[\log \frac{\tilde{p}(z_1, z_2)}{p_\theta(z_1)p_\theta(z_2)}] \tag{80}$$

$$= \mathbb{E}_{p_\theta(z_1, z_2)}[\log \frac{p_\theta(z_1, z_2)}{\tilde{p}(z_1, z_2)}] \tag{81}$$

$$= D_{\mathrm{KL}}(p_\theta(z_1, z_2) \parallel \tilde{p}(z_1, z_2)) \tag{82}$$

Due to the non-negative property of KL divergence, it can be obtained that:

$$D_{\mathrm{KL}}(p_\theta(z_1, z_2) \parallel p_\theta(z_1)p_\theta(z_2)) - (\mathbb{E}_{p_\theta(z_1, z_2)}[f] - \log \phi) \geq 0 \tag{83}$$

$$D_{\mathrm{KL}}(p_\theta(z_1, z_2) \parallel p_\theta(z_1)p_\theta(z_2)) \geq (\mathbb{E}_{p_\theta(z_1, z_2)}[f] - \log \phi) \tag{84}$$

$$\geq (\mathbb{E}_{p_\theta(z_1, z_2)}[f] - \log \mathbb{E}_{p_\theta(z_1)p_\theta(z_2)}[e^f]) \tag{85}$$

Therefore, we can obtain:

$$\max I(Z_1; Z_2) \propto \max \mathbb{E}_{p_\theta(z_1, z_2)}[f] - \log \mathbb{E}_{p_\theta(z_1)p_\theta(z_2)}[e^f] \tag{86}$$

Based on the above derivation, the Proposition 5 can be proved.

$\square$

