# OpenReview forum: "Representation Matching Information Bottleneck for Text Matching in Asymmetrical Domains"
_ICLR.cc/2024/Conference — Submitted to ICLR 2024_

### Official Review · Reviewer_fokd · 2023-10-29

**Soundness:** 3 good
**Presentation:** 3 good
**Contribution:** 2 fair
**Rating:** 6
**Confidence:** 2

**Summary:**

The paper addresses the issue of aligning text representations in asymmetrical domains to improve text matching performance. It introduces RMIB framework, which narrows the distribution of text representations and emphasizes the importance of interaction between text representations. The paper theoretically proves that optimizing the RMIB is equivalent to maximizing the mutual information between text representations given the task information. The contributions include proposing RMIB, providing its theoretical foundation, and demonstrating its effectiveness on various text matching models and datasets.

**Strengths:**

The paper presents an extensive experimental evaluation, including detailed results across various datasets, demonstrating the significant improvements achieved through the RMIB framework. The inclusion of a range of evaluation metrics adds robustness to the assessment of the RMIB framework's performance in diffferent text matching tasks. The paper offers a strong theoretical foundation, showcasing the equivalence between text representation distribution narrowing and information bottleneck optimization, thereby reinforcing the validity and relevance of the proposed RMIB framework.

**Weaknesses:**

How the authors performed statistical significant test for Table 2? The analysis primarily focuses on scenarios with limited data availability, and the paper lacks a comprehensive exploration of the RMIB framework's effectiveness in highly heterogeneous data matching scenarios. And also the need for manual hyperparameter tuning within the RMIB optimization process might restrict its applicability to diverse text matching tasks. Future research should aim to automate the hyperparameter selection process to enhance the framework's adaptability and scalability. While the results demonstrate the effectiveness of the RMIB framework, further comparisons with state-of-the-art models on a broader set of text matching tasks would strengthen the paper's conclusions and provide a more comprehensive understanding of the framework's capabilities and limitations.

**Questions:**

How the authors performed statistical significant test for Table 2?

---

> ### Author Response · Authors · 2023-11-14
> **Rebuttal by Authors**
>
> Thank you very much for your careful review. We try our best to answer your questions.
>
> ***
>
> Q1: How the authors performed statistical significant test for Table 2?
>
> A: In order to ensure that the improvement in experimental results is due to RMIB and not other factors, such as random initialization, learning rate schedule, model parameters, etc., we use the same environment and experimental settings for both baseline and RMIB. At the same time, in order to avoid the randomization error caused by parameter initialization, we used the same seed in all experiments to ensure the reproducibility of results. This is mentioned in section "IMPLEMENTATION DETAILS".
>
> ***
>
> Q2: The analysis primarily focuses on scenarios with limited data availability, and the paper lacks a comprehensive exploration of the RMIB framework's effectiveness in highly heterogeneous data matching scenarios. And also the need for manual hyperparameter tuning within the RMIB optimization process might restrict its applicability to diverse text matching tasks. Future research should aim to automate the hyperparameter selection process to enhance the framework's adaptability and scalability. While the results demonstrate the effectiveness of the RMIB framework, further comparisons with state-of-the-art models on a broader set of text matching tasks would strengthen the paper's conclusions and provide a more comprehensive understanding of the framework's capabilities and limitations.
>
> A: Our paper mainly focuses on asymmetric domain matching, and in future work, we will try to extend RMIB in multimodal data matching. We have mentioned this in section 'CONCLUSION AND LIMITED'. We have also mentioned hyperparameter tuning during the RMIB optimization process in this section. To the best of my knowledge, DDR-Match is currently the best model in asymmetric domain text matching. We have compared it in Table 2.

---

### Official Review · Reviewer_jW8Q · 2023-10-31

**Soundness:** 3 good
**Presentation:** 3 good
**Contribution:** 3 good
**Rating:** 6
**Confidence:** 4

**Summary:**

This paper investigates text-matching tasks in asymmetrical domains from the perspective of information bottleneck theory. It demonstrates that narrowing the distribution of text representations in asymmetrical domains text matching is equivalent to optimizing the information bottleneck. Furthermore, it extends the conventional information bottleneck approach to a novel framework called Representation Matching Information Bottleneck (RMIB). The theoretical justification of the proposed RMIB method is provided, and empirical evidence is presented to show its effectiveness in improving model performance for text retrieval and matching tasks.

**Strengths:**

1. The proposed RMIB method offers a reasonable improvement to the information bottleneck approach by considering the practical aspects of text-matching tasks. It captures unique factors within text-matching tasks, such as the interaction between text representations and the limitations of a single text representation. These ideas demonstrate a certain level of innovation.
2. The methods presented in the paper are accompanied by clear theoretical proofs.
3. Building on the theoretical analysis, the effectiveness of RMIB is further validated through empirical experiments. Special cases in the experiments are also analyzed and explained.
4. The paper exhibits a well-structured hierarchy and a clear line of thought, making it highly readable.

**Weaknesses:**

1. The paper contains errors in the tables presenting experimental results. While the experiments are described as "F1 values on SICK," the tables do not include the F1 metric for SICK. Additionally, based on the information provided in "3.1 DATESET AND METRIC," it seems that the dataset metrics don't mention the F1 score.
2. The proof for Proposition 4 in the paper is somewhat perplexing, and the meaning and proof process for Theorem 4 are not well understood. Moreover, there is an error in Equation (67) within the proof of Proposition 4.
3. The method needs to set three hyperparameters, which could be limiting in practical applications. The author acknowledges this limitation in the paper.

**Questions:**

1. According to "3.1 DATASET AND METRIC," why do SICK and SciTail have F1 metrics, and why do the tables of experimental results not align with the F1 metric as described in the experiments?
2. Regarding the proof of Theorem 4: Can we directly derive equation (68) based on a Markov chain, and why are equations (66) and (67) necessary? My knowledge in this field is limited, so I seek your understanding if I have misunderstood.
3. How does equation (68) directly lead to the conclusion, and can this step be explained in more detail?
4. I'm not quite sure how to implement the objective function using code. How do I calculate the KL divergence between a Gaussian distribution and the distribution of text representations?

---

> ### Author Response · Authors · 2023-11-14
> **Rebuttal by Authors**
>
> Thank you very much for your careful review. We try our best to answer your questions.
>
> ***
>
> W1: The paper contains errors in the tables presenting experimental results. While the experiments are described as "F1 values on SICK," ... the F1 score. Q1: According to "3.1 DATASET AND METRIC," why do SICK and SciTail have F1 metrics, and why do the tables of experimental results not align with the F1 metric as described in the experiments?
>
> A: This is a typo. We have modified 'F1 values on SICK' to 'accuracy on SICK'
>
> ***
> W2: The proof for Proposition 4 in the paper is somewhat perplexing, and the meaning and proof process for Theorem 4 are not well understood. Moreover, there is an error in Equation (67) within the proof of Proposition 4.
>
> A: There is a typo in Eq(67). The $I(Y ;Z_1, Z_2|Y )$ should be $I(Y ;Z_1, Z_2|Z )$, but the proof process is correct. In the previous line of Eq (68), we used  $I(Y ;Z_1, Z_2|Z )$ instead of $I(Y ;Z_1, Z_2|Y )$. We sincerely thank you for conducting such a meticulous review. The typos you pointed out are of great help in improving the quality of the paper.
>
> ***
>
> Q2: Regarding the proof of Theorem 4: Can we directly derive equation (68) based on a Markov chain, and why are equations (66) and (67) necessary? My knowledge in this field is limited, so I seek your understanding if I have misunderstood. Q3: How does equation (68) directly lead to the conclusion, and can this step be explained in more detail?
>
> A: According to the definition of mutual information: $I(Y;Z_1,Z_2,Z)=I(Y;Z_1, Z_2)+I(Y;Z|Z_1,Z_2)=I(Y;Z)+I(Y;Z_1,Z_2|Z)$. Since $Y\to (X_1,X_2)\to(Z_1,Z_2)\to Z$ forms a Markov chain, $Y$ and $Z$ are independent of each other given $(Z_1,Z_2)$, that is, $I(Y;Z|Z_1,Z_2)=0$, then:
> $$
> \begin{equation}
>  I(Y;Z_1, Z_2)+0 =I(Y;Z)+I(Y;Z_1,Z_2|Z)
>  \end{equation}
> $$
>
> Based on the non-negativity of mutual information, then: $ I(Y;Z_1,Z_2)\ge I(Y;Z) $. We can see that $I(Y;Z_1,Z_2)$ is the upper bound of $I(Y;Z)$.
> Therefore, if the cost of optimizing $I(Y; Z_1, Z_2)$ and $I(Y; Z)$ to the same value is the same, this means that optimizing $I(Y; Z)$ will result in a higher $I(Y; Z_1, Z_2)$. Since one of the optimization goals of RMIB is to maximize $I(Y; Z_1, Z_2)$, the higher the value of $I(Y; Z_1, Z_2)$, the better $Z_1^\*,Z_2^\*$ learned.
>
>
> ***
>
> Q4: I'm not quite sure how to implement the objective function using code. How do I calculate the KL divergence between a Gaussian distribution and the distribution of text representations?
>
> A: As we mentioned in Section 3.2, we use the reparameter trick to optimize KL terms in RMIB. Taking BERT as an example, the pseudocode of its feedforward calculation is:
> ```
> # let x1 and x2 are the input texts
> x1_token_embedding, x2_token_embedding=BERT(x1,x2)
>
> # maxpooling to get the text embedding
> x1_embedding=max_pooling(x1_token_embedding)
> x2_embedding=max_pooling(x2_token_embedding)
>
> # using reparameter trick
> x1_mean=mlp_mean(x1_embedding)
> x1_std=mlp_std(x1_embedding)
> x1_repara_embedding=x1_mean + normal(mean=0, std=1)*x1_std
>
> x2_mean=mlp_mean(x2_embedding)
> x2_std=mlp_std(x2_embedding)
> x2_repara_embedding=x2_mean + normal(mean=0, std=1)*x2_std
>
> #concat text embedding
> x=concat([x1_repara_embedding, x2_repara_embedding])
>
> # get the output
> logit=mlp(x)
>
> # get class prob
> class_prob=softmax(logit)
> ```
>
> After reprocessing the text representation with reparameter trick, `x1_repara_embedding` can be represented as sampled from the distribution `normal(mean=x1_mean, std=x1_std)`. Therefore, we can optimize the KL item in RMIB:
> $$ KL[p_\theta(x1\\_repara\\_embedding|x_1,x_2)\parallel p(\mathcal{Z})] \propto 0.5 * [ square(x_1\\_mean)+ square(x_1\\_std)-\log( square(x_1\\_std))-1 ]    $$ where $p(\mathcal{Z})$ is the standard normal distribution.

---

> > ### Comment · Reviewer_jW8Q · 2023-11-23
> >
> > Thank you for your reply which answers my questions. I will keep my score.

---

### Official Review · Reviewer_4HVN · 2023-11-02

**Soundness:** 2 fair
**Presentation:** 3 good
**Contribution:** 2 fair
**Rating:** 5
**Confidence:** 4

**Summary:**

In this paper, the authors propose to align representations of texts from asymmetric domains for better matching performance. Specifically, the authors leverage the information theory to show the alignment solution is not only narrowing the distributions, but also equivalent to optimizing the information bottleneck.  Several proofs are also given to support the proposed ideas. Experiments on several benchmark datasets demonstrate that the proposed method outperforms the previous work DDR-MATCH. An ablation study also shows that it is beneficial to add both interaction and inadequacy to information bottleneck.

**Strengths:**

* S1: The paper has strong theoretical supports by having proofs from the view of information theory.
* S2: Experiments demonstrate the significant gains over the baseline with different embedding models.

**Weaknesses:**

* W1: While most of the method descriptions are in the form of information theory, the actual model architecture deployed in the experiment should be also clarified. Some comparisons to baseline methods should be also conducted, otherwise we might not know if the gain is from the proposed idea or simply because of other factors like more model parameters.
* W2: Lack of comparisons to other representation alignment methods, such as [a,b,c,d]
* W3: With the same encoder, the representations are to some degree still in the same domain. The real asymmetric setup (like [d]) with different encoders or even different data types should be considered in the experiments.

--
[a] Imani, E., Hu, W., & White, M. (2022). Representation Alignment in Neural Networks. Transactions on Machine Learning Research.
[b] Bjerva, J., Kouw, W., & Augenstein, I. (2019, September). Back to the future–sequential alignment of text representations. In Proceedings of the 34th AAAI Conference on Artificial Intelligence.
[c] Wang, T., & Isola, P. (2020, November). Understanding contrastive representation learning through alignment and uniformity on the hypersphere. In International Conference on Machine Learning (pp. 9929-9939). PMLR.
[d] Duan, J., Chen, L., Tran, S., Yang, J., Xu, Y., Zeng, B., & Chilimbi, T. (2022). Multi-modal alignment using representation codebook. In Proceedings of the IEEE/CVF Conference on Computer Vision and Pattern Recognition (pp. 15651-15660).

**Questions:**

* Q1: Many gains are huge in terms of values. I wonder if the authors conduct significance tests to have verification.
* Q2: Following W2, I wonder if the authors can compare with more representation alignment methods during the author feedback period.
* Q3: Following W3, the proposed method actually does not use any property about text, so theoretically it can be applied in representations of arbitrary data formats. I wonder if there could be some experiments on multi-modal settings.
* Q4: In Table 1, it is interesting that the proposed method improves a lot in `SICK`, but also significantly underperform DDR-Match in `SNLI`. I wonder if the authors have conducted analysis to research this phenomenon.

**Details Of Ethics Concerns:**

N/A.

---

> ### Author Response · Authors · 2023-11-14
> **Rebuttal by Authors - Part 1**
>
> Thank you very much for your careful review. We try our best to answer your questions.
>
> ***
>
> W1: While most of the method descriptions are ... like more model parameters.
>
> A: In fact, RMIB did not propose a new model structure. RMIB constrains the learned representations on existing models to better handle asymmetric domains text matching task, so we have not provided too detailed descriptions of these models. We have uploaded the main implementation code for this paper in Supplementary Material on how to implement the RMIB. To ensure the effectiveness of RMIB, we also replicated the baseline model in our experimental environment to ensure fairness in comparison. From the experimental results, it can be seen that in the scenario of asymmetric domains text matching, these baselines do not perform as well as RMIB. To the best of my knowledge, DDR-Match is currently the best model in this scenario. We have also added it to Table 2 for comparison. In addition to cross entropy loss, RMIB also includes three other items. Therefore, we also conducted ablation experiments to verify the overall effectiveness of RMIB. The results in Table 3 validate the effectiveness of RMIB.
>
> ***
>
> W2: Lack of comparisons to other representation alignment methods, such as [a,b,c,d].
> Q2: Following W2, I wonder if the authors can compare with more representation alignment methods during the author feedback period.
>
> A: Our paper focuses on the task of asymmetric domains text matching rather than representation alignment task. The 4 papers you provided are not directly related to asymmetric domains text matching. In the works related to asymmetric domains text matching, such as DDR Mach, these papers have not compared with these methods. After carefully reading these 4 papers, we find that these methods are not suitable for asymmetric domains text matching. The following is the corresponding explanation of the reasons.
>
> The representation matching in asymmetric domains text matching means that the global distribution of text representations from different domains in the common semantic space should be as similar as possible, so as to capture the commonality between potential text representations.
>
> [a] explored the effectiveness of neural network transfer. On the regression task (UCI CT Position Dataset) and image classification task (Cifar10 and MNIST), it was found that after training, the top singular vector of the neural network was aligned to the target. They also explained why aligning the top singular vector with the target can accelerate learning. However, they did not propose any new representation matching methods.
>
> [b] found Changes in language usage cause a data drift between time-steps and some way of controlling for the shift between time-steps is necessary. For example, BERT rarely appeared in scientific papers before 2018. After the publication of BERT, however, usage has increased in frequency. They apply a domain adaptation technique to correct for shifts.  The sequential alignment of text representations referred to in [b] is completely different from the representation matching in asymmetric domains. Meanwhile, the task targeted by [b] is sequence annotation or sequence prediction, which is not the same as asymmetric domain text matching. As mentioned in Section "Experimental Setup" in [b]: "As we are dealing with dynamical learning, the vast majority of NLP data sets can unfortunately not be used since they do not include time stamps.".
>
> [c] proposed an contrastive learning method. Representation alignment in contrastive learning refers to pulling in samples with similar semantics in the semantic vector space and pushing out samples with significant semantic differences. The representation alignment here is not the same as the representation matching in asymmetric domains. Let ($x$, $x^{+}$) are semantically similar sentences, while ($x$, $x^{-}$) are semantically opposite sentences. The contrastive learning objective is $D (f (x), f (x^{+}))>D (f (x), f (x^{-}))$, where $D$ is the distance measurement formula and $f$ is the text encoding model. In asymmetric domain matching tasks, we need to consider the actual task information.  For example, in the information retrieval task, let $x$ and $x^{+}$are two similar queries, then $x^{-}$ corresponds to doc, and our goal is not $D (f (x), f (x^{+}))>D (f (x), f (x^{-}))$,, but $\max p (y | x, x ^{-})$, where $y$ is the task label for $x$ and $x^{-}$. From the perspective of task format, the goal of contrastive learning is to ensure that sentence representations with similar semantics are as close as possible, while our task is to predict the target semantic labels. Therefore, this method cannot be directly applied to asymmetric text matching tasks.
>
> [d] explored image and text matching tasks. The setting of this task is completely different from asymmetric domain text matching, so this method cannot be directly applied to asymmetric text matching tasks.

---

> ### Author Response · Authors · 2023-11-14
> **Rebuttal by Authors - Part 2**
>
> W3: With the same encoder, the representations are to some degree still in the same domain. The real asymmetric setup (like [d]) with different encoders or even different data types should be considered in the experiments.
>
> A: In asymmetric domain text matching tasks, asymmetric means that the texts to be matched are sampled from two different domains. For example, in the information retrieval tasks, the query and the retrieved document clearly belong to different domains. Therefore, for asymmetric text matching task, when the texts to be matched are obtained from different domains, it constitutes a real asymmetric setup, regardless of whether you use the same decoder or not. There are still certain differences between multimodal data matching tasks and asymmetric domain based matching tasks. Our current paper mainly focuses on asymmetric domain based matching. In future work, we will attempt to expand RMIB in multimodal data matching scenarios, as mentioned in section "CONFUSION AND LIMITED LIMITED".
>
> ***
>
> Q1: Many gains are huge in terms of values. I wonder if the authors conduct significance tests to have verification.
>
> A: In order to ensure that the improvement in experimental results is due to RMIB and not other factors, such as random initialization, learning rate schedule, model parameters, etc., we use the same environment and experimental settings for both baseline and RMIB. At the same time, in order to avoid the randomization error caused by parameter initialization, we used the same seed in all experiments to ensure the reproducibility of results. This is mentioned in section "IMPLEMENTATION DETAILS".
>
> ***
>
> Q3: Following W3, the proposed method actually does not use any property about text, so theoretically it can be applied in representations of arbitrary data formats. I wonder if there could be some experiments on multi-modal settings.
>
> A: For asymmetric text matching tasks, there are currently two types of models: text encoding model and text interaction model. When using these two models, we are actually using some properties of the text. For example, for text interaction model, we can concatenate the two asymmetric domain texts to be matched as a whole for processing. However, for multimodal tasks, we cannot directly concatenate images and text as a whole representation. Our current work mainly focuses on the asymmetric domain text matching. We will attempt to expand RMIB in multimodal data matching scenarios, as mentioned in section "CONFUSION AND LIMITED LIMITED".
>
> ***
>
> Q4: In Table 1, it is interesting that the proposed method improves a lot in SICK, but also significantly underperform DDR-Match in SNLI. I wonder if the authors have conducted analysis to research this phenomenon.
>
> A: Compared to SNLI, we have noticed a more significant improvement in RMIB in SICK. We think this is because SNLI have significantly more data than SICK. For example, SNLI has about 57 times the data than SICK. [1] indicates that when the number of training data sets increases, the generalization performance of the model will also increase. When there are enough training sets, the model itself has good generalization performance, so RMIB can bring less improvement. We have analyzed and explained this phenomenon in the section "PERFORMANCE OF RMIB".
>
> [1] Vladimir Koltchinskii. Rademacher penalties and structural risk minimization. IEEE Transactions on Information Theory, 47(5):1902–1914, 200

---

> > ### Comment · Reviewer_4HVN · 2023-11-23
> >
> > I acknowledge that I have read all of the author responses. As some of my concerns have been addressed, I increased my score to 5.

---

### Official Review · Reviewer_xd94 · 2023-11-06

**Soundness:** 3 good
**Presentation:** 3 good
**Contribution:** 2 fair
**Rating:** 6
**Confidence:** 4

**Summary:**

Authors apply information bottleneck theory in asymmetric text matching to improve latent text code quality. The challenge in asymmetric text matching is finding the mapping from two distinct text distributions (e.g. questions and answers) to a common latent vector space.

**Strengths:**

Clear experiments and results.

**Weaknesses:**

weaknesses are adequately discussed by the authors.

**Questions:**

In the Interaction, we maximize I(z1;z2). This makes sense for positive and negative examples, but do we want to maximize I(z1;z2|Y=neutral)?

---

> ### Author Response · Authors · 2023-11-14
> **Rebuttal by Authors**
>
> Thank you very much for your careful review. We try our best to answer your questions.
>
> ***
>
> Q1: In the Interaction, we maximize I(z1;z2). This makes sense for positive and negative examples, but do we want to maximize I(z1;z2|Y=neutral)?
>
> A1: In fact, if we only view $I (z_1; z_2)$ separately without considering task label information, its effectiveness is limited. For example, assuming ($x_1$, $x_2$) are positive while ($x_1$, $x_3$) are negative, this means that $x_1$ should interact as much as possible with $x_2$ and $x_3$, but the semantic labels of ($x_1$, $x_2$) and ($x_1$, $x_3$) are completely different. Therefore, we believe that the information of task labels needs to be taken into account when conducting representational interactions. $I (z_1; z_2 | Y)$ refers to the assumption that the model can correctly predict labels, and the interaction between representations should be as much as possible. This is not related to whether the labels are positive, negative or neutral, but rather to whether the model can correctly predict label.

---

### Author Response · Authors · 2023-11-20
**Looking forward to the reviewer's reply**

Dear ACs,

We look forward to feedback from the reviewers. We addressed all reviewers' concerns very carefully by providing evidence. In addition, we believe that the misunderstandings from the reviewers led to some concerns, and we did our best to justify these misunderstandings in the responses. Therefore, we are eager to discuss with the reviewers, which can also help refine our manuscript. If reviewers have any questions, we are open to continuing the discussion.

At last, we thank all reviewers and ACs for their time and efforts.

Best,

---

### Meta-Review · Area_Chair_kXNa · 2023-12-07

**Metareview:**

This paper addresses the problem of matching texts across domains. The intuition is that for this kind of task, representations should be domain-invariant. They set out to explicitly achieve this by aligning texts to a shared prior. The paper argues for a number of formal properties of the proposed bottleneck method (some which contained typos, now fixed).

Perhaps the major weakness here is the empirical setup. As pointed out by reviewer 4HVN, the authors do not compare against existing text matching strategies (they respond that existing methods are not appropriate for the "asymmetric" scenario, perhaps; though elaboration here would have been welcome), and consider only a single encoder, limiting the implication of the work.

**Justification For Why Not Higher Score:**

My sense is that this is a solid piece of technical work but will be of only modest interest to the ICLR community, especially in light of the limited experimental setup.

**Justification For Why Not Lower Score:**

N/A

---

### Decision · Program_Chairs · 2024-01-16

Reject